# Geometric Analysis of Token Selection in Multi-Head Attention

## Abstract

We present a geometric framework for analysing multi–head attention in large language models (LLMs). Without altering the mechanism, we view *standard* attention through a top-$N$ selection lens and study its behaviour directly in value–state space. We define geometric metrics—*Precision*, *Recall*, and *F-score*—to quantify separability between selected and non–selected tokens, and derive non–asymptotic bounds with explicit dependence on dimension and margin under empirically motivated assumptions (stable value norms with a compressed sink token, exponential similarity decay, and piecewise attention–weight profiles). The theory predicts a small–$N$ operating regime of strongest non–trivial separability and clarifies how sequence length and sink similarity shape the metrics. Empirically, across LLaMA-2-7B, Gemma-7B, and Mistral-7B, measurements closely track the theoretical envelopes: top-$N$ selection sharpens separability, sink similarity correlates with *Recall*. We also found that in LLaMA-2-7B heads specialise into three regimes—*Retriever*, *Mixer*, and *Reset*—with distinct geometric signatures. Overall, attention behaves as a structured geometric classifier with measurable criteria for token selection, offering head–level interpretability and informing geometry–aware sparsification and design of attention in LLMs.

## 1 Introduction

Attention is a context-dependent weighting mechanism that enables a model to aggregate information from short-term history without compressing it into a fixed bottleneck Bahdanau et al. (2016). Scaled up within the *Transformer* architecture Vaswani et al. (2023), stacked self-attention and feed-forward layers have largely supplanted recurrence and convolutions, forming the core of modern large language models (LLMs).

A growing theoretical literature studies the mathematical properties of attention in Transformers. Results establish expressive power, including universal approximation with positional encodings Yun et al. (2020); Perez et al. (2021); Bhattamishra et al. (2020); Mudarisov et al. (2025). A complementary line of work interprets self-attention as a normalised kernel smoother, motivating efficient linear variants with controlled approximation error Choromanski et al. (2022).

Geometry offers complementary views of attention. First, attention coefficients live on the probability simplex $\Delta^{n-1}$: softmax can be seen as the solution to an entropy-regularised optimisation, while sparse alternatives (sparsemax, entmax) induce structured, low-dimensional solutions Martins & Astudillo (2016); Peters et al. (2019). Second, geometry also shapes the representation space: rotary positional encodings act as rotations that encode relative phase Su et al. (2023), and hyperbolic variants adapt similarity to curved spaces to model hierarchical structure Ganea et al. (2018).

Recent empirical studies have revealed extreme-token phenomena such as the *attention sink*, an emergent and apparently universal effect across LLMs Gu et al. (2025), and the specialisation of heads into active and dormant roles Guo et al. (2024). Other approaches propose explicit token-selection strategies—e.g., *OrthoRank*, which leverages orthogonality to the sink token as a ranking criterion Shin et al. (2025). While these works highlight the importance of geometry and selection in attention, they stop short of a unified theoretical framework that quantifies geometric separability.

**Main idea: view multi-head attention as a *classifier in value–state space*.** We model multi-head attention as a classifier operating in the value–state space. Under the hypothesis that attention

aggregates the top-$N$ relevant representations while minimising interference from the remaining context, we propose selection schemes and define *Precision* and *Recall* directly in value–state space to quantify separability between selected and non-selected tokens. Under empirically grounded assumptions, we derive bounds on the expected metrics that depend on both dimensionality and margin. We validate the theory by measuring these quantities across layers and heads in open LLMs.

**Our contributions.**

1. **Geometric classifier view & metrics.** We cast standard multi–head attention as a classifier in value–state space under a top-$N$ selection lens, and define *Precision*, *Recall*, and *F-score* with interpretable radii ($r_{\min}, r_{\max}$) to quantify separability.

2. **Theory with empirically grounded assumptions.** Assuming (i) stable value norms with a compressed sink, (ii) exponential similarity decay, and (iii) piecewise attention–weight profiles, we derive non–asymptotic, dimension– and margin–dependent bounds for the metrics, predict a small–$N$ operating regime for strongest separability, and show that sink similarity chiefly biases *Recall*.

3. **Head taxonomy & validation.** We identify three robust regimes—*Retriever*, *Mixer*, *Reset*—with functional hypotheses (retrieve near $v_L$, mix $v_L$ with relevant context, reset via near–orthogonal components), provide depth–wise distributions and simple labelling criteria, and validate predictions for LLMs.

## 2 PROBLEM SETTING

We consider the classical multi-head attention mechanism Vaswani et al. (2023). Let $X \in \mathbb{R}^{L \times d}$ denote the matrix of token representations with rows $x_i^\top$. Queries, keys, and values are obtained via linear projections $Q = XW^Q$, $K = XW^K$, $V = XW^V$. For a single head, attention computes a weighted sum of values:

$$\text{Attn}(Q, K, V) = \text{softmax}\left(\frac{QK^\top}{\sqrt{d_k}}\right) V,$$

and multi-head outputs are concatenated and linearly projected (via $W^O$). This construction yields a row-stochastic attention matrix, placing attention weights on the probability simplex and thereby inducing a non-trivial geometry over token embeddings.

While full-token aggregation is effective, prior work shows it can collect diffuse or redundant information Brunner et al. (2020); Choromanski et al. (2022); Katharopoulos et al. (2020). In our study we analyse the *selection* induced by the top-$N$ attention weights and quantify separability in value–state space. Given attention weights $\{\alpha_i\}_{i=0}^{L}$ and corresponding value vectors $\{v_i\}_{i=0}^{L}$, let $I_N$ denote the indices of the top-$N$ tokens by attention weight. We define the *representative vector* as

$$s = \sum_{i \in I_N} \alpha_i\, v_i. \tag{1}$$

This induces an inner geometry over tokens in which each head and layer performs information-based selection. Selection-based aggregation is conceptually related to sparse and structured attention mechanisms Martins & Astudillo (2016); Peters et al. (2019); Niculae & Blondel (2019), as well as recent empirical studies on sink tokens and token filtering Shin et al. (2025); Gu et al. (2025); Guo et al. (2024). Unlike these approaches, our focus is the *geometric separability* of the selected tokens, which we quantify using adapted classification metrics and analyse both theoretically and empirically.

## 3 THEORY

We focus on the geometric properties of multi-head attention. Consider attention weights $\{\alpha_i\}_{i=0}^{L}$ and corresponding value-state embeddings $\{v_i\}_{i=0}^{L}$ for a fixed head and layer. The standard aggregation constructs the representative vector, i.e. a weighted average of all value states using the attention distribution.

To study selection-based aggregation, we assume the model selects the indices of the top-$N$ tokens $I_N \subseteq \{0, \dots, L\}$, where $1 \leq N \leq L$. We aim to quantify how well the selected tokens separate from the non-selected ones in the value-state space. To this end, we adapt classical classification metrics to the geometric setting.

**Definition 1.** *Let $B_r(s)$ be a ball of radius $r$ centered at $s$. The **Precision** of selection $I_N$ is defined as*

$$P(r, N) = \frac{|\{\alpha_i v_i \in B_r(s) | i \in I_N\}|}{|\{\alpha_i v_i \in B_r(s) | i \in I_N\}| + |\{\alpha_i v_i \in B_r(s) | i \notin I_N\}|} \quad (2)$$

*It measures the fraction of tokens near $s$ that belong to the selected set.*

**Definition 2.** *Let $B_r(s)$ be a ball of radius $r$ centered at $s$. The **Recall** of selection $I_N$ is defined as*

$$R(r, N) = \frac{|\{\alpha_i v_i \in B_r(s) | i \in I_N\}|}{N} \quad (3)$$

*It measures the fraction of selected tokens that lie near $s$.*

Since the full dependence on $r$ is analytically intractable, we consider two extremal choices:

1. $r_{\min} = \min\limits_{i \notin I_N} \|\alpha_i v_i - s\|_2$, i.e. minimal distance to the non-selected token.

2. $r_{\max} = \max\limits_{i \in I_N} \|\alpha_i v_i - s\|_2$, i.e. maximum distance to the selected token.

By construction,

$$P(r_{\min}, N) \equiv 1, \quad R(r_{\max}, N) \equiv 1. \quad (4)$$

These radii provide interpretable bounds that connect geometric separability to selection quality.

Figure 1 illustrates the intuition in a 2D toy example: tokens lie on a circle, some selected and others not. If normalisation is imperfect, a selected token may lie far from $s$, reducing Precision and Recall. In the example, $\mathrm{Precision} = 3/8$ and $\mathrm{Recall} = 2/3$.

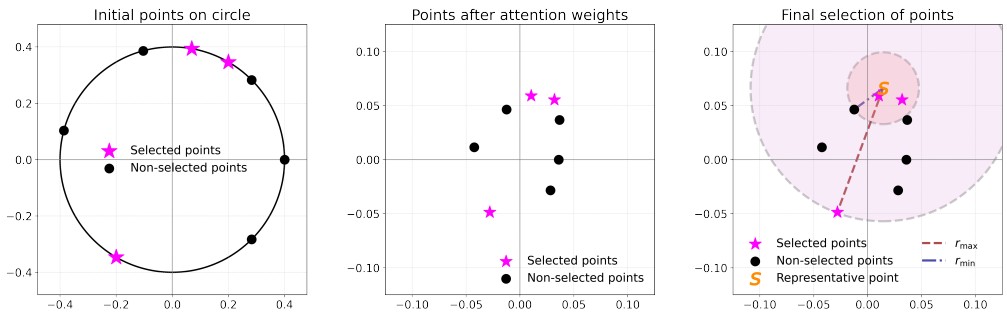

Figure 1: **Illustrative 2D example of geometric separability in value–state space. Left:** Token embeddings lie on a circle. **Middle:** After scaling by their attention weights $\alpha_i$, both selected (magenta stars) and non-selected (black dots) points move toward the origin. **Right:** We see the points lying in the $B_{r_{\min}}(s)$ and $B_{r_{\max}}(s)$ as well as the $r_{\min}$ and $r_{\max}$.

Continuing the intuition behind the classification model, we can consider the F-measure:

**Definition 3.** *Given Precision and Recall, the **F-score** is their harmonic mean:*

$$F_s(r, N) = \frac{2P(r, N)R(r, N)}{P(r, N) + R(r, N)}. \quad (5)$$

Before turning to theorems, we state the assumptions underlying our analysis.

### 3.1 NORMS OF VALUE STATES

We analyse geometric features of attention through the norms of value–state embeddings across heads and layers. Empirically, two consistent patterns emerge. First, as shown in Figure 2(left), the

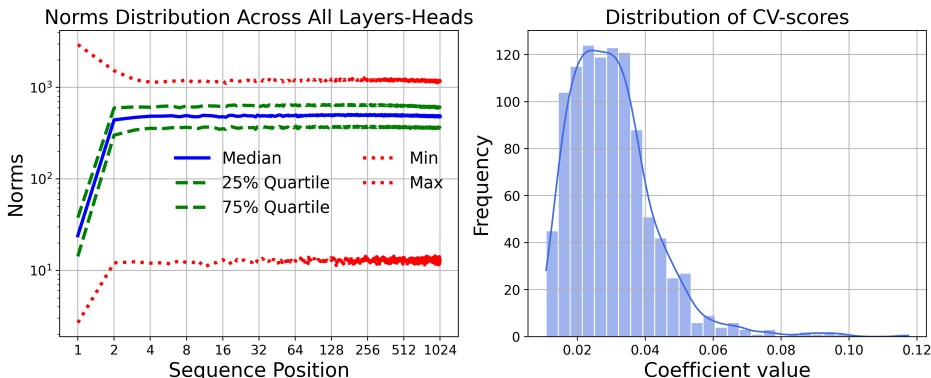

Figure 2: **Except attention sink value–state norms remain stable. Left:** Average norm of value states across all layer–head pairs. **Right:** Coefficient of variation of the non-sink norms.

first position exhibits a markedly smaller norm than all subsequent positions, reflecting the well-known *attention sink*. Beyond this, token norms rapidly stabilise and remain nearly constant across positions $i = 0, \ldots, 1024$, with tight concentration around the median.

Second, the right panel of Figure 2 reports the distribution of coefficient-of-variation (CV) scores across layer–head pairs. Most values lie in a narrow range around $0.02$–$0.03$, with only a small fraction exceeding $0.05$. This concentration at low CV values indicates that the norms $\|v_i^{(l,h)}\|_2$ vary very little and are effectively constant across the majority of layers and heads.

Motivated by these observations, we introduce the following assumption.

**Assumption 1 (Value–state norms).** For each layer $l$ and head $h$, value–state vectors satisfy:

1. **Stable norms:** For tokens $i > 0$,
$$\|v_i^{(l,h)}\|_2 = C_{l,h} \quad \text{for some } C_{l,h} > 0.$$

2. **Sink-token norm:** For the first token $(i = 0)$,
$$\|v_0^{(l,h)}\|_2 = \lambda \, C_{l,h}, \qquad 0 < \lambda < 1, \;\; \lambda \ll 1.$$

This assumption places non-sink value states on subspheres of radius $C_{l,h}$, with the sink token compressed towards the origin by a factor $\lambda$.

## 3.2 CROSS TOKEN SIMILARITY

We now turn to the geometric structure of cross-token similarity. Given value-state vectors $\{v_i\}_{i=0}^{L}$, we measure their pairwise cosine similarity:

$$\cos(v_i, v_j) = \langle v_i, v_j \rangle / \|v_i\|_2 \, \|v_j\|_2, \tag{6}$$

where $\langle \cdot, \cdot \rangle$ denotes the standard dot product.

Figure 3 shows the distribution of MAE values across layer-heads. Most heads fall within $0.05$–$0.10$ ($5\% - 10\%$ error), indicating a good exponential fit, while a smaller subset reaches $0.20$–$0.25$, suggesting deviations from the simple exponential law (up to $25\%$ error).

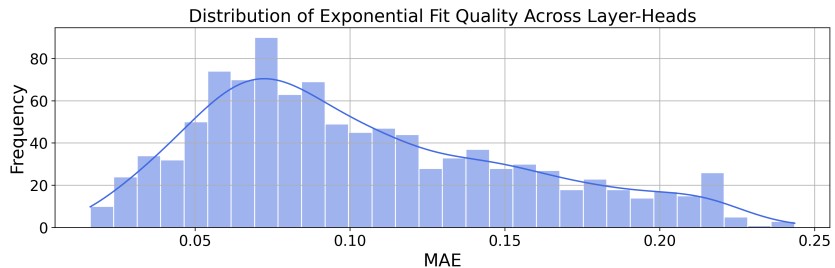

Figure 3: **Exponential fitting provides a good approximation with low error.** The distribution of mean absolute error across layer-heads shows the average 9% error.

> **Assumption 2 (Cross-token similarity).** For each layer $l$ and head $h$:
>
> 1. **Exponential decay.** For tokens $i, j > 0$,
>
> $$\cos(v_i^{(l,h)}, v_j^{(l,h)}) \;=\; e^{-\beta|i-j|},$$
>
> where $\beta > 0$ is a sensitivity parameter.
>
> 2. **Sink connections.** For the first token $i = 0$,
>
> $$\cos(v_0^{(l,h)}, v_j^{(l,h)}) \;=\; \rho_0, \quad \forall j \neq 0.$$

### 3.3 WEIGHTS FUNCTIONAL DEPENDENCE

Since value-state norms remain nearly constant across tokens, the variability of attention is primarily encoded in the attention weights themselves. Figure 4 shows that the distribution of attention mass follows a consistent four-phase structure: (i) an *attention sink* at position 0 with probabilities two orders of magnitude larger than content tokens; (ii) a *constant plateau* over positions 1–100, with uniform mass around $10^{-5}$; (iii) an *oscillatory regime* across positions 100–800, where fluctuations resemble sinusoidal patterns and may encode syntactic or semantic cycles; and (iv) an *exponential recency bias* in the final positions (800–1024), where probabilities grow rapidly. This structure – sink, plateau, oscillation, and recency – shows that attention distributions are far from uniform, instead reflecting complex mathematical regularities that encode linguistic structure and positional dependencies.

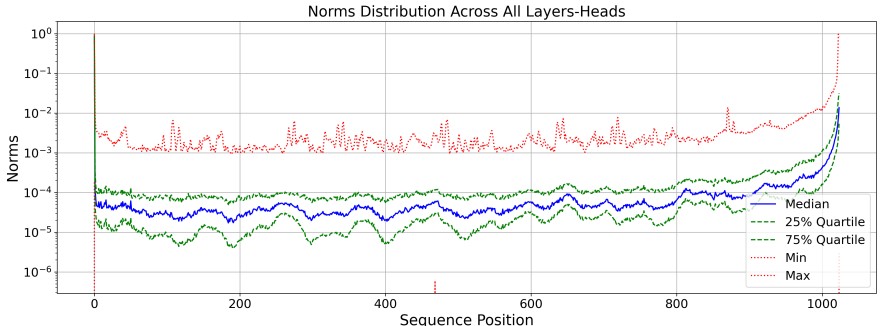

Figure 4: **Attention mass follows a piecewise-structured distribution across sequence positions.** This piecewise structure highlights that attention is highly non-uniform, reflecting positional and linguistic regularities.

Therefore, we propose the following assumption:

**Assumption 3 (Attention Weights).** The attention distribution follows:

$$\alpha_i = \begin{cases} p_{\text{sink}} & \text{if } i = 0 \\ p_{\text{base}} & \text{if } 1 \le i \le T_1 \\ p_{\text{base}} \cdot (1 + \eta \cdot \cos(\omega \cdot i)) & \text{if } T_1 < i \le T_2 \\ p_{\text{base}} \cdot e^{\eta \cdot (i - T_2)} & \text{if } i > T_2 \end{cases}$$

**Parameters:**

- $p_{\text{sink}}$ (sink probability)
- $\eta$ (sensitivity parameter for both oscillation and growth)
- $\omega$ (frequency rate)
- $T_1$ (end of constant phase)
- $T_2$ (start of exponential phase)

**Note.** In practice it may happen that $T_1 = T_2$, in which case the oscillatory phase is absent.

## 3.4 THEORETICAL ANALYSIS

We analyze the behavior of the metrics defined in (2)–(3) under Assumptions 1–3. Throughout, write $a_i := \alpha_i, \|v_i\|_2$ and denote cosine similarities by $\rho_{jk} := \cos(v_j, v_k)$. Let:

$$S_{\text{in}} = \sum_{k \in I_N \setminus \{0\}} a_k, \quad a_{\min}^{\text{in}} = \min_{i \in I_N \setminus \{0\}} a_i,$$

$$a_{\max}^{\text{out}} = \max_{j \notin I_N} a_j, \quad \rho_{\text{out}} = \max_{\substack{j \notin I_N \\ k \in I_N \setminus \{0\}}} \rho_{jk} \le e^{-\beta}. \tag{7}$$

We then introduce the *margin parameter* and the scale factor:

$$\Delta = (a_{\min}^{\text{in}})^2 - 2 a_{\max}^{\text{out}} S_{\text{in}} \rho_{\text{out}}, \quad \Delta_0 = a_0^2 - (S_{\text{in}} \rho_{\text{in}})^2. \tag{8}$$
$$B = a_{\max}^{\text{in}} + a_{\max}^{\text{out}} + S_{\text{in}},$$

For the upper bound we define

$$p_{-ij} = \begin{cases} \phi(\xi_{i,j})/(\xi_{i,j} + 1/\xi_{i,j}), & \xi_{i,j} \ge 0, \\ 1/2, & \xi_{i,j} < 0. \end{cases} \tag{9}$$

where $\xi_{i,j} = \mu_{i,j}/\sigma_{i,j}$, with

$$\mu_{i,j} = (a_j^2 - a_i^2) - 2 a_j M_j + 2 a_i M_i, \quad \sigma_{i,j} = B/\sqrt{2\kappa d}, \tag{10}$$

and

$$M_j = \sum_{l \in I_N \setminus \{0\}} a_l \rho_{l,j}. \tag{11}$$

**Theorem 1** (Precision bound). *Assume $\Delta > 0$ and let $\kappa > 0$ be the concentration constant. Then*

$$\mathbb{E} P(r_{\max}, N) \gtrsim \left( 1 + (L - N) e^{-\kappa d (\Delta/B)^2} + \frac{L-N}{N} e^{-\kappa d (\Delta_0/B)^2} \right)^{-1}, \tag{12}$$

$$\mathbb{E} P(r_{\max}, N) \lesssim 1 - \frac{1}{N+1} \max_{i \in I_N, j \notin I_N} p_{-ij}. \tag{13}$$

Intuitively, Precision remains close to 1 when the margin $\Delta$ is positive and the dimension $d$ is large.

**Theorem 2** (Recall bound). *Under the same assumptions,*

$$\mathbb{E} R(r_{\min}, N) \ge 1 - (L - N) e^{-\kappa d (\Delta/B)^2} - \frac{L-N}{N} e^{-\kappa d (\Delta_0/B)^2}, \tag{14}$$

$$\mathbb{E} R(r_{\min}, N) \lesssim \frac{1}{N} \sum_{i \in I_N} \left( 1 - \max_{j \notin I_N} \underline{p}_{ij} \right). \tag{15}$$

Thus, Recall approaches 1 provided the effective margin is sufficiently large relative to the scale $B$.

**Corollary 1** (F-score bounds). *Under the same assumptions,*

$$\frac{2R_{\min}(r_{\min}, N)}{1 + R_{\min}(r_{\min}, N)} \leq \mathbb{E}F(r_{\min}, N) \leq \frac{2R_{\max}(r_{\min}, N)}{1 + R_{\max}(r_{\min}, N)}, \tag{16}$$

$$\frac{2P_{\min}(r_{\max}, N)}{1 + P_{\min}(r_{\max}, N)} \leq \mathbb{E}F(r_{\max}, N) \leq \frac{2P_{\max}(r_{\max}, N)}{1 + P_{\max}(r_{\max}, N)}. \tag{17}$$

To proceed with the theorems analysis, we propose consideration of the Appendix A.2.

## 4 EXPERIMENTS

We validate the theoretical analysis on `LLaMA-2-7B` with context length 1024 using OpenWeb-Text Gokaslan et al. (2019). Our goal is to assess how the bounds from Theorems 1–2 align with empirical behaviour.

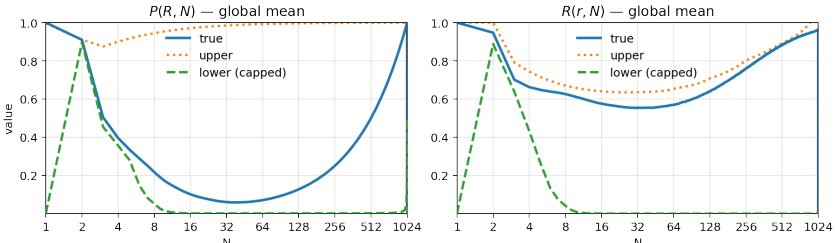

Figure 5: **Attention achieves its strongest non-trivial separability at small selection sizes ($N \in [1, 4]$), while separability is weakest at intermediate $N$.** Global means of *Precision* (left) and *Recall* (right) versus $N$ (solid blue), with theoretical upper (dotted orange) and lower (dashed green, capped at 0) envelopes.

Figure 5 reports global averages of *Precision* (left) and *Recall* (right) against the number of selected tokens $N$. The empirical curves lie within the theoretical envelopes predicted by Theorems 1–2. Upper bounds (orange) are tight in the small–$N$ and large–$N$ regimes and remain close to the empirical Recall for moderate $N$. Lower bounds (green) are conservative—especially for mid–$N$—but capture the rise near $N{=}1$–2 and the vanishing floor as $N$ grows, consistent with the margin– and dimension–dependent terms.

These results support the geometric classifier view: separability, as measured in value–state space, is maximised when either a few highly salient tokens are selected or when the selection is nearly exhaustive, and it is minimised at intermediate $N$ where the margin between selected and non–selected tokens is smallest. The agreement between empirical trends and our bounds indicates that the margin/scale quantities derived from Assumptions 1–3 reliably predict how selection size impacts Precision and Recall.

Figure 5 shows attention behaving like a margin–based classifier in value–state space: separability (Precision/Recall) is highest for small selections and degrades at intermediate $N$. Selecting only a few tokens ($N{=}1$–2) yields a large effective margin because top–weight tokens cluster near the representative point; as $N$ moves into the mid–range (tens), borderline tokens are admitted, the margin shrinks, and separability drops. For very large $N$ both metrics return to 1 trivially as almost all tokens are selected. Hence, the practical operating regime that maximises non-trivial separability is *small $N$*, near the "elbow" before the sharp Precision decline (typically $N \in [1, 4]$ on our models), with larger $N$ used only to meet Recall targets.

### 4.1 ATTENTION SINK EFFECT

We next examine the geometric effect of the attention–sink token by testing the predicted link between its similarity $\rho_0$ and *Recall*. Figure 6 reports the correlation $\mathrm{corr}\big(R(r_{\min}, N), \rho_0\big)$ as a function of $N$ across models.

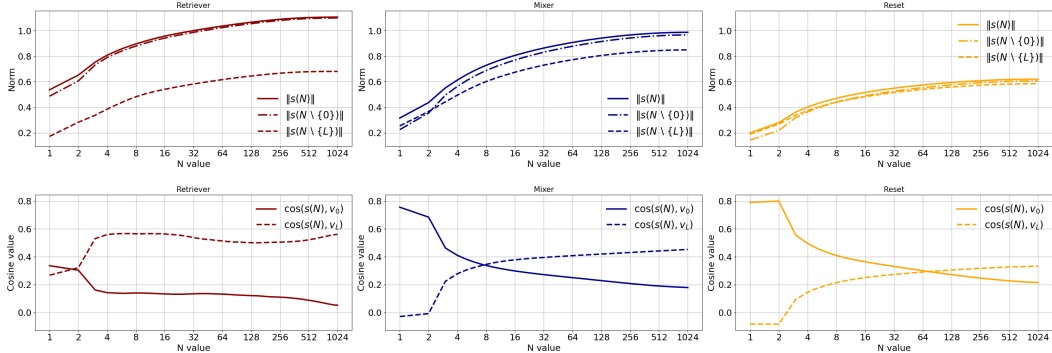

Figure 7: **Geometric interpretation highlights distinct mechanisms of head regimes.** Top: norms of $s(N)$ (solid) and leave-one-out variants removing the sink (dash–dot) or last token (dashed). Bottom: cosine similarities with $v_0$ (solid) and $v_L$ (dashed). From left to right: *Retriever*: contributions reinforce a stable direction anchored to $v_L$. *Mixer*: alignment shifts from the sink to the last token as $N$ grows, consistent with gating/combination. *Reset*: the sink anchors the representation with weak $v_L$ alignment, and information is accumulated gradually from multiple directions.

A consistent pattern emerges: correlations are *strongest at small $N$*, peaking around $N \approx 3$ for Gemma and Mistral ($\sim 0.6$), while LLaMA rises more gradually from near zero to $\sim 0.25$ by $N \approx 6$. As $N$ enters the mid–range (tens), correlations decline (Gemma) or plateau (Mistral), reflecting reduced separability when borderline tokens are added. For large $N$ (hundreds), correlations climb again and stabilise ($\sim 0.4$–$0.45$ for LLaMA/Mistral; $\sim 0.3$ for Gemma) before a mild downturn at the extreme end, where selection is nearly exhaustive.

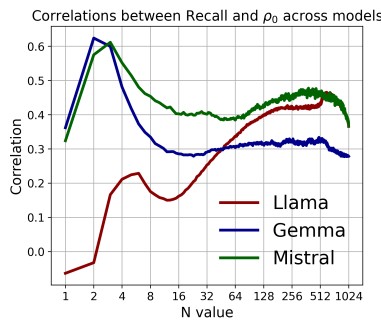

Figure 6: **Sign of the $\rho_0$ impact the Recall statistics.** Correlations peak around $N \approx 3$ (Gemma/Mistral $\approx 0.6$).

Higher $\rho_0$ increases the chance that tokens near the sink are co–selected when $N$ is small, boosting Recall; the effect weakens as $N$ admits lower–margin tokens and strengthens again when $N$ is large enough that coverage dominates. Thus, the sink's similarity is a reliable predictor of Recall in the regime where separability is meaningful (small $N$), with model–specific strength.

Overall Figure 6 shows that the average cosine similarity $\cos(v_0, v_j)$ (for $j \neq 0$) is strongly correlated with the average Recall score across $N$. The sink token exerts a *systematic geometric influence* on attention separability, improving Precision when its direction aligns with other embeddings.

### 4.2 GEOMETRIC HEAD TAXONOMY

We classify heads by how they balance the sink token ($v_0$), the current/last token ($v_L$), and the remaining tokens, using the relative magnitudes of $\|\alpha_0 v_0\|_2$, $\|\alpha_L v_L\|_2$, and the aggregate of the rest, together with how the representative vector $s(N)$ aligns with $v_0$ and $v_L$ as $N$ grows:

- **Retriever.** Last token dominates: $\|\alpha_L v_L\|_2$ is largest; $s(N)$ stays strongly aligned with $v_L$ (high $\cos(s, v_L)$) across $N$.

- **Mixer.** Last token is intermediate between sink and rest: alignment shifts from $v_0$ at small $N$ towards $v_L$ as $N$ increases (decreasing $\cos(s, v_0)$, increasing $\cos(s, v_L)$).

- **Reset.** Sink dominates or mass is broadly spread: $\|\alpha_0 v_0\|_2$ is comparable to or exceeds the others; $s(N)$ remains sink–biased with weak $\cos(s, v_L)$ except at very large $N$.

Figure 7 provides diagnostics: the norm of $s(N)$ and leave–one–out variants $s(N \setminus \{0\})$, $s(N \setminus \{L\})$ (top), and cosine alignment with $v_0$ and $v_L$ (bottom).

*Retriever* heads show a sharp drop in $\|s(N)\|$ when removing $L$ and consistently high $\cos(s, v_L)$ thus retrieving representations similar to the current token, and acting as local copy/nearest–neighbour modules.

*Mixer* heads act as routers: $\cos(s, v_0)$ is high at small $N$ but decreases as $\cos(s, v_L)$ increases, and the two leave–one–out norms converge with $N$. This mixes the current token with other relevant tokens, behaving like a gated convex combination that shifts from sink to content as $N$ increases.

*Reset* heads are sink–biased: removing the sink reduces $\|s(N)\|$ more than removing $L$; $\cos(s, v_L)$ stays low until very large $N$. This replaces the current token by injecting components largely orthogonal to $v_L$, akin to a normalising/reset operation.

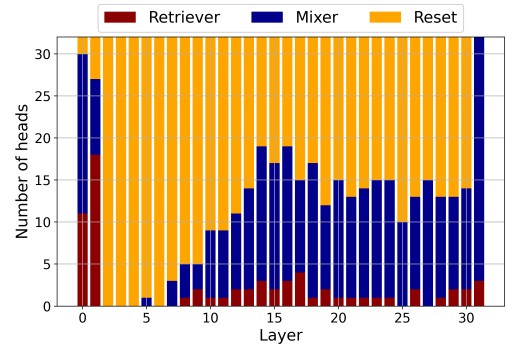

Figure 8: **Distribution of the regimes across layers.** The figure dynamics of the head regimes in Llama-2-7B model.

Figure 8 shows how the taxonomy varies with depth. *Retriever* heads are most visible at the first two layers, aligning with an intuitive function of early aggregation of maximally similar representations. In the first few layers, *Reset* (sink–biased) heads prevail, consistent with a normalising/initial re-encoding role that suppresses idiosyncratic token features and injects common components. From the lower–middle layers onwards, *Mixer* heads become predominant, indicating increased combination of the current token with context—precisely where our geometric analysis predicts moderate, stable separability. Overall, the progression *Retriever* → *Reset* → *Mixer* across depth suggests a pipeline in which models first stabilise representations, then integrate contextual evidence, and finally refine token-aligned features for prediction.

Examples of value norms for these three head types and regime–wise precision–recall curves are provided in Appendix A.4.

# 5 CONCLUSION

We presented a geometric framework for analysing multi-head attention in large language models (LLMs). Viewing *standard* attention through a top-$N$ selection lens (without changing the mechanism), we defined Precision, Recall, and F-score in value–state space and derived bounds with explicit dependence on dimension and margin under empirically motivated assumptions (stable norms, exponentially decaying similarities, and piecewise attention-weight profiles). The resulting predictions align closely with measurements on LLaMA-2, Gemma, and Mistral.

Our analysis yields three takeaways. First, value norms are highly stable across positions, cross-token similarity is well modelled by exponential decay, and attention weights follow a consistent four-phase pattern (sink, plateau, oscillation, recency). Second, the sink token exerts a systematic geometric effect: its alignment correlates with Recall and shapes separability of the selected set. Third, attention heads specialise into *Retriever*, *Mixer*, and *Reset* regimes, each exhibiting characteristic precision–recall trade-offs that our theory explains.

Together, these results suggest that attention behaves as a structured geometric classifier with measurable criteria for token selection, and that similar geometric regularities recur across model families. This perspective opens avenues for geometry-aware sparsification, diagnostics for head functionality, and interpretable architectural or training interventions in large-scale language modelling.

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

## A  APPENDIX

### A.1  PROOFS OF THE THEOREMS

**Notation.**  Fix a layer–head pair and $N \in \{1, \ldots, L\}$. Let $I_N$ be the indices of the top–$N$ weights. Let's Retriever:

$$u_i = v_i/\|v_i\|_2, a_i = \alpha_i \|v_i\|_2, y_i = a_i u_i, s = \sum_{k \in I_N} y_k,$$

$$D_\ell = \|s - y_\ell\|_2^2, r_{\max} = \max_{i \in I_N} D_i^{1/2}, r_{\min} = \min_{j \notin I_N} D_j^{1/2} \tag{18}$$

We study

$$P(r_{\max}, N) = N/(N + K_{\text{out}}),$$

with $K_{\text{out}} = \sum_{j \notin I_N} \mathbf{1}\{D_j \le r_{\max}^2\}$

$$R(r_{\min}, N) = K_{\text{in}}/N,$$

with $K_{\text{in}} = \sum_{i \in I_N} \mathbf{1}\{D_i \le r_{\min}^2\}$.

**Deterministic inequalities (key reductions).**  Recall

$$r_{\max}^2 = \max_{i \in I_N} D_i, \qquad r_{\min}^2 = \min_{j \notin I_N} D_j, \tag{19}$$

$$K_{\text{out}} = \sum_{j \notin I_N} \mathbf{1}\{D_j \le r_{\max}^2\}, \qquad K_{\text{in}} = \sum_{i \in I_N} \mathbf{1}\{D_i \le r_{\min}^2\}. \tag{20}$$

If $D_j \le r_{\max}^2 = \max_{i \in I_N} D_i$, then there exists $i \in I_N$ with $D_j \le D_i$. Hence, for each $j \notin I_N$,

$$\boxed{\mathbf{1}\{D_j \le r_{\max}^2\} \le \sum_{i \in I_N} \mathbf{1}\{D_j \le D_i\}}. \tag{21}$$

Summing over $j \notin I_N$ yields

$$\boxed{K_{\text{out}} \le \sum_{j \notin I_N} \sum_{i \in I_N} \mathbf{1}\{D_j \le D_i\}}. \tag{22}$$

Similarly, if $D_i > r_{\min}^2 = \min_{j \notin I_N} D_j$, then there exists $j \notin I_N$ with $D_i \ge D_j$. Therefore, for each $i \in I_N$,

$$\boxed{\mathbf{1}\{D_i > r_{\min}^2\} \le \sum_{j \notin I_N} \mathbf{1}\{D_i \ge D_j\} = \sum_{j \notin I_N} \mathbf{1}\{D_j \le D_i\}}. \tag{23}$$

Summing over $i \in I_N$ gives

$$\boxed{N - K_{\text{in}} = \sum_{i \in I_N} \mathbf{1}\{D_i > r_{\min}^2\} \le \sum_{i \in I_N} \sum_{j \notin I_N} \mathbf{1}\{D_j \le D_i\}}. \tag{24}$$

These purely deterministic bounds reduce both precision and recall to double sums of pairwise comparisons $\mathbf{1}\{D_j \le D_i\}$.

**Pairwise tail bound (with highlighted margins and tails).** Let $X_{ij} = D_j - D_i$. Using $D_\ell = \|s\|_2^2 + \|y_\ell\|_2^2 - 2\langle s, y_\ell \rangle$ and $s = \sum_{k \in I_N} y_k$, we obtain

$$\boxed{X_{ij} = (a_j^2 - a_i^2) - 2 \sum_{k \in I_N} \langle y_k, \ y_j - y_i \rangle}. \tag{25}$$

For large enough $d$, $\langle u_a, u_b \rangle$ are subgaussian with variance proxy $O(1/d)$ and that $\rho_{jk} \leq \rho_{\text{out}}$ for $j \notin I_N$, $k \in I_N \setminus \{0\}$. Define

$$S'_{\text{in}} = \sum_{k \in I_N \setminus \{0\}} a_k, \qquad a_{\min}^{\text{in}} = \min_{i \in I_N \setminus \{0\}} a_i, \qquad a_{\max}^{\text{out}} = \max_{j \notin I_N} a_j, \tag{26}$$

and the scale

$$B = a_{\max}^{\text{in}} + a_{\max}^{\text{out}} + S'_{\text{in}}. \tag{27}$$

Then the mean gaps (margins) satisfy

$$\boxed{\mu_{ij} := \mathbb{E}X_{ij} \ \geq \ \Delta := (a_{\min}^{\text{in}})^2 - 2\,a_{\max}^{\text{out}} S'_{\text{in}} \rho_{\text{out}} \quad (i \neq 0),} \tag{28}$$

and for the sink index $i = 0$

$$\boxed{\mu_{0j} \ \geq \ \Delta'_0 := a_0^2 - \left(S'_{\text{in}} \rho_{\text{out}}\right)^2.} \tag{29}$$

Moreover $X_{ij}$ is subgaussian with variance proxy

$$\boxed{\sigma^2 \ \leq \ c\,\frac{B^2}{d},} \tag{30}$$

for a universal constant $c > 0$. Hence Chernoff tails give, for some universal $\kappa > 0$,

$$\boxed{\mathbb{P}\{X_{ij} \leq 0\} \ \leq \ \exp\{-\kappa\,d\,(\Delta/B)^2\}, \quad \mathbb{P}\{X_{0j} \leq 0\} \ \leq \ \exp\{-\kappa\,d\,(\Delta'_0/B)^2\}.} \tag{31}$$

Denoting $p_{ij} = \mathbb{P}\{D_j \leq D_i\}$, the same bounds apply to $p_{ij}$.

Now we can easily obtain the bounds, using the deterministic bounds, Jensen inequality and tail estimations.

## A.2 Theorem's analysis

**Effect of parameters.** We next analyze the qualitative effect of model parameters on the derived bounds. Table 1 summarizes their roles and impacts.

| Parameter | Role (sink-aware setup) | Effect on lower bounds ($\uparrow$ improves, $\downarrow$ weakens) |
|---|---|---|
| $d$ (dimension) | Concentration factor in $e^{-\kappa d (\cdot/B)^2}$ | $\uparrow$ exponentially (larger $d \Rightarrow$ smaller tails) |
| $L$ (length) | Union factors $(L-N)$ and $(L-N)/N$ in bounds | $\downarrow$ linearly as $L$ grows |
| $N$ (top-$N$) | Defines $I_N$; changes union factors and geometry | Mixed: $(L-N)$ terms $\downarrow$ (improves), but $S'_{\text{in}}$ $\uparrow$, $a_{\min}^{\text{in}} \downarrow$ so $\Delta$, $\Delta_0$ may weaken |
| $a_0$ (sink) | Enters $\Delta_0 = a_0^2 - (S'_{\text{in}} \rho_{\text{out}})^2$ | $\uparrow$ (often strong gain, esp. for $EP(r_{\max}, N)$) |
| $a_{\min}^{\text{in}}$ | Smallest inside effective weight | $\uparrow$ (larger margin $\Delta$) |
| $a_{\max}^{\text{out}}$ | Largest outside effective weight | $\uparrow$ when smaller (reduces penalty term in $\Delta$) |
| $\beta$ (decay of $\rho$) | Controls $\rho_{ij} \approx e^{-\beta|i-j|}$ | $\uparrow$ (larger $\beta \Rightarrow$ smaller $\rho_{\text{out}}$) |
| Profile $(\alpha_i)$ via $(\eta, \omega, T_1, T_2)$ | Shapes $a_{\min}^{\text{in}}, a_{\max}^{\text{out}}, S'$ | Peaks entering $I_N$: $\uparrow$; valleys entering $I_N$: $\downarrow$ |

Table 1: Influence of parameters on the lower bounds for $\mathbb{E}P(r_{\max}, N)$ and $\mathbb{E}R(r_{\min}, N)$ (and hence on $\mathbb{E}F_s$).

**Dependence on $N$.** The theorems assume fixed $N$, but the effective set $I_N$ varies with $N$. Table 2 illustrates five regimes.

| Regime | Composition of $I_N$ | Trend of $\Delta/B$ | Effect on statistics |
|---|---|---|---|
| I: $N = 1$ | $\{0\}$ only | n/a | All statistics equal to 1 |
| II: $1 < N < m_1$ | $\{0\}$ + top of tail | $\uparrow$ (large $\alpha_{(N)}$, small $S_{\text{in}}$) | $\Delta > 0$ and large $\Rightarrow$ bounds near 1 |
| III: $m_1 \leq N < m_2$ | Add oscillatory peaks | $\searrow$ ($\alpha_{(N)} \downarrow$, $S_{\text{in}} \uparrow$) | Still high; slow decline. Larger $\eta, \omega$ help. |
| IV: $m_2 \leq N < m_3$ | Valleys + $1..T_1$ in | $\Downarrow$ ($\alpha_{(N)} \approx p_{\text{base}}$; jump in $S_{\text{in}}$) | $\Delta$ often $\leq 0$; bounds weaken |
| V: $N \geq m_3$ | All oscillations + tail | $\downarrow$ (small $\alpha_{(N)}$, large $S_{\text{in}}$) | Weak guarantees; require large $d$ or $\beta$ |

Table 2: Qualitative dependence of the bounds on the parameter $N$.

**Effect of the sink.** Finally, we analyze the role of the sink token through $\rho_0$. For the Recall metric we can write

$$R(r, N) = \frac{1}{N} \sum_{i \in I_N} \mathbf{1}\{\|s - \alpha_i v_i\|_2 \leq r\} = F_{\text{in}}(r), \tag{32}$$

where $F_{\text{in}}$ is a CDF. For squared distances $d_i^2 = \|\alpha_i v_i - s\|_2^2$ we obtain $d_i(\text{neg}) \approx d_i(\text{ort}) + \frac{a_i S_{\text{in}} \rho_0}{d_i(\text{ort})}$, so that the shift in the minimal radius satisfies $\Delta r_{\min} = r_{\min}(\text{ort}) - r_{\min}(\text{neg})$. Hence

$$\Delta R(r, N) \approx F'(r_{\min}(\text{ort})) \Delta r < 0, \tag{33}$$

since $\rho_0 < 0$.

*Interpretation.* Negative sink correlations push non-selected tokens away from the representative vector $s$, reducing separation.

### A.3 GEOMETRICAL REPRESENTATION

Figure 9 provides a geometric visualization of the multi-head attention mechanism, where each head is represented as a spherical subspace in three dimensions. The spheres capture how different attention heads project the input representations into distinct subspaces, thereby enabling the model to capture diverse contextual relationships. This spherical abstraction emphasizes the complementary nature of multiple heads, illustrating how they jointly span different regions of the representation space to enrich feature extraction.

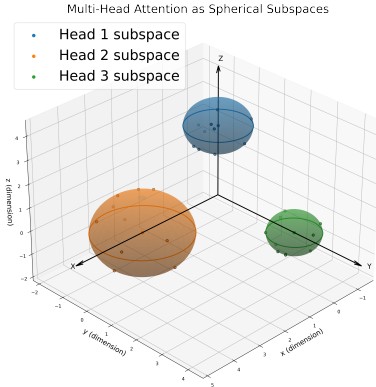

Figure 9: Representation of multi-head spherical geometry of the attention mechanism.

### A.4 PRECISION–RECALL BY HEAD REGIME

For completeness, Figure 11 compares Precision and Recall across the three regimes defined in the main text. Differences are modest in global means, consistent with Figure 7 showing that regime effects are most visible in geometry (alignment and leave-one-out norms) rather than in raw PR levels.

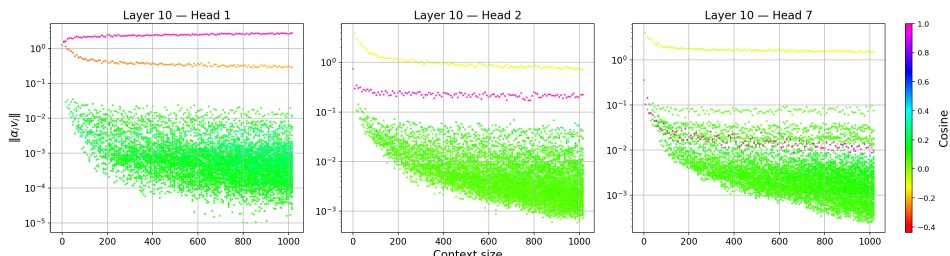

Figure 10: **Attention heads exhibit distinct operational regimes.** Scatter of contributions $\|\alpha_i v_i\|_2$ across context sizes (colour: $\cos(v_i, v_j)$). *Retriever* heads (left) are last-token dominated; *Mixer* heads (centre) place the last token between sink and content; *Reset* heads (right) spread mass broadly, with no single dominant last token.

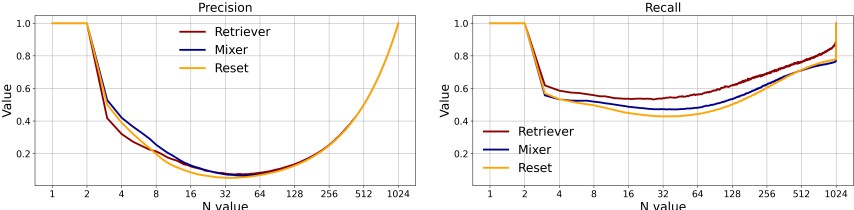

Figure 11: **Head regimes show complementary precision–recall trade-offs (appendix).** *Retriever* achieves relatively higher recall at large $N$ but lower precision; *Mixer* maintains moderate recall with gradually decreasing precision; *Reset* exhibits a U-shaped profile, with both metrics improving again at large $N$.

