# OpenReview forum: "Geometric Analysis of Token Selection in Multi-Head Attention"
_ICLR.cc/2026/Conference — Submitted to ICLR 2026_

### Official Review · Reviewer_uLa2 · 2025-10-24

**Soundness:** 2
**Presentation:** 1
**Contribution:** 2
**Rating:** 2
**Confidence:** 3

**Summary:**

This paper provides a geometric interpretation of self-attention by viewing it as a separator/classifier operating in the value space. The authors propose to evaluate each attention head’s behavior through precision, recall, and F-score metrics that quantify separability between the selected (top-N) and non-selected value tokens. Under a set of assumptions about value-vector norms, similarity decay, and attention profiles, they analytically predict that attention heads operate in a small-N regime where separability is maximized. Empirically, they identify three broad types of heads (Retriever, Mixer, Reset) and show how their prevalence varies with model depth.

**Strengths:**

- The theoretical framing is novel and gives an interesting geometric perspective on attention, with clear analytical predictions that can be empirically checked.
- The observation that attention sinks are not mere no-ops, but instead play an active role (especially in Recall and Reset-type heads), is particularly interesting and could motivate deeper investigation into sink dynamics and normalization mechanisms in large models.

**Weaknesses:**

- The paper is not particularly well presented. Key motivations are unclear: the authors do not sufficiently explain why attention should be modeled as a classifier or why geometric separability in value space is the right lens for interpretability. The classification framework feels somewhat imposed rather than naturally derived from prior literature or empirical necessity.
- Several important concepts (e.g. the “MAE” mentioned in the text) are never properly defined or justified in the context of their framework.
- The paper does not make a compelling case for why separability, precision, or recall are meaningful or diagnostic of downstream behavior, model efficiency, or interpretability.
- Certain claims (such as an “oscillatory regime” across positions 100–800, or “semantic cycles”) are presented without quantitative backing or follow-up experiments. These explanations offered without proof weaken the empirical credibility of the work.
- The experiments are limited to a small set of 7B-parameter models and do not explore robustness across architectures, tasks, or long-context settings. Given the theoretical emphasis, a more comprehensive experimental grounding is needed.
- Given the current level of analysis and the modest experimental depth, the paper would be better suited for a workshop rather than a full-conference publication.

**Questions:**

- In Barbero et al., 2025, the authors explicitly discuss the formation of sharp heads through the influence of attention sinks. Do the authors believe that this mechanism might be related to the findings in the paper? Hows is the top-N selection scheme related to head sharpness?
- How are separability metrics expected to evolve in long-context settings?

Barbero, Federico, et al. "Why do LLMs attend to the first token?." arXiv preprint arXiv:2504.02732 (2025).

---

> ### Author Response · Authors · 2025-11-19
> **Response to reviewer uLa2**
>
> Dear reviewer,
>
> Thank you for your review and for the helpful comments.
>
> Weaknesses:
>
> 1. The goal of our paper is not to claim that attention is a supervised classifier, but that each head behaves like a margin-based selector in value space. Each head outputs a representative vector $s$, a weighted combination of value vectors. Top-$N$ tokens are our probe of how much information a small subset of tokens carries about this representative. We then ask: given $s(N)$, how well does the mechanism geometrically separate the selected tokens from the rest? We study metric-based separability in Euclidean space, which is natural since attention outputs are linear combinations of value vectors. Furthermore, we focus on value space because it is the representation propagated to the next layer and is robust to rescaling of queries/keys that changes logits, but not the geometry of values.
>
> 2. We partially agree here. To save space, we did not repeat explicit MAE formulas in later figure captions, assuming the reader would transfer the logic from Section 3.1 to 3.2. In the revised version, we will restate how MAE is computed and report it directly in the relevant captions.
>
> 3. You are right that we do not connect our geometric metrics directly to downstream quantities such as perplexity or task loss. The aim of this work is to develop and validate a mathematical model, not to provide a full downstream evaluation. We formulate several metric questions (how to define separability, how to bound it, which parameters matter) and answer them within the classical transformer setup, with empirical validation on existing models. Studying downstream implications (e.g., how separability relates to perplexity, alignment, or robustness) is an important direction, but combining a full theoretical analysis with a broad downstream study is beyond the scope of this paper.
>
> 4. Sections 3.1–3.3 are intended to justify the theoretical assumptions and limitations. In 3.1 and 3.2 we introduce the assumptions and provide quantitative validation; in 3.3 we give qualitative interpretations of patterns such as the “oscillatory regime” or “semantic cycles”. These phrases are informal descriptions that help relate Assumption 3 to Figure 4, rather than additional formal assumptions. In the revision, we will clarify this explanatory role and moderate the language around “semantic cycles” to emphasize that it is a qualitative interpretation, not a claim about all heads.
>
> 5. In our experiments, we use LLaMA-2-7B and similarly sized Mistral and Gemma models. Due to computational constraints (all experiments on a single 16GB GPU), we were not able to systematically evaluate larger models. Regarding robustness across architectures, tasks, and long contexts: (a) we focus on models with similar transformer-style architectures and setups, so we do not include fundamentally different architectures; (b) we address the metric/geometric problem (Section 2): defining and estimating separability, understanding which parameters influence it, and examining head taxonomies within a single framework; (c) we study different context lengths when analyzing head taxonomies, since this part depends on $L$, while the geometric bounds and the sink connection are stated for general $L$.
>
> Questions:
>
> 1. We are familiar with Barbero et al. and appreciate their contribution. Our analysis focuses on the geometric behavior of attention and independently arrives at complementary ideas. (a) Our Reset heads can be viewed as geometric realizations of an anti-over-mixing mechanism: the sink acts as an anchor that keeps the representative vector $s$ away from a collapsing subspace, similar in spirit to their mechanism that prevents excessive mixing. (b) Regarding head sharpness: in Barbero et al., “sharp” heads place most of their mass on a small number of tokens. This is close to our top-$N$ perspective. For a sharp head, using small $N$ in top-$N$ selection captures almost all attention mass, and our Precision/Recall metrics then quantify how consistently those few important tokens cluster around the head’s output in value space. In short, sharpness describes how many tokens matter; our separability metrics describe how coherently those tokens align with what the head writes into the residual stream.
>
> 2. Our theory suggests two opposing effects as context length $L$ increases: (a) larger $L$ increases the size of the non-selected set; this appears in our bounds and can weaken separability, especially for larger $N$; (b) mechanisms that keep heads sharp (including sink effects highlighted by Barbero et al.) preserve a large effective margin for the few selected tokens, maintaining high Precision/Recall for small $N$. Therefore, for long-context models we expect non-trivial separability to remain maximized in a small-$N$ regime, while absolute Precision/Recall levels may decrease for heads that become more mixing as $L$ grows.
>
> Thank you again for your careful and insightful review.

---

### Official Review · Reviewer_9XZ2 · 2025-10-31

**Soundness:** 3
**Presentation:** 2
**Contribution:** 3
**Rating:** 4
**Confidence:** 3

**Summary:**

This paper attempts to understand multi-head attention through a novel view of token selection. The authors provide a geometric analysis based on separability between selected and non-selected tokens. To conduct this, they provide a theoretical framework: first make assumptions on the norm states, token similarities, attention dynamics, and then derive the bounds for precision/recall/F1-score of separability of selected/non-selected tokens. Finally, based on the built geometric framework, the authors aim to interpret the head functions.

**Strengths:**

The authors provide a novel view of multi-head attention with nice theoretical support. Using the geometric perspective, they interpret the head functions in multi-head attention. The findings on value norms (assumption 1) are interesting.

**Weaknesses:**

My major concerns towards this paper are about the validity of several assumptions in this paper, and the possible actionable suggestions on the LLM/attention community.

1. About assumption 2, I am a little bit concerned whether it makes sense to use an exponential function (between 0 and 1) to model a cosine similarity (ranged from -1 to 1). Although the authors show that the MAE error is very small, I am wondering whether it makes sense to assume that the cosine similarity could be always non-negative. I am looking forward to more empirical evidence.

2. About assumption 3, the attention dynamics in Figure 4 follow my intuition that the first token is an attention sink and the recent tokens typically have higher attention. However, such a simplification may ignore existence of induction heads [1], or some other sink tokens in middle context [2]. And it does not make sense to me that all heads/layers follow such a pattern, especially considering the authors fail to show some fitting errors like in assumption 2.

3. When we discuss about multi-head attention, normally we are considering attention as a matrix. However, here it seems that authors mainly discuss the attention weights on the final token. I am wondering whether the conclusions are still valid when $L$ is largely different.

4. Although the authors claim that the LLaMA, Gemma, Mistral follow the theoretical analysis on bounds, I suggest to show that the previous assumptions also hold in these models or different sizes (other than 7B), or different data domains.

5. I acknowledge that the theoretical framework in this paper is elegant, please clarify how this research can provide actionable suggestions/insights to the LLM/attention community.

Some minors:

1. Please clarify the model/data used in assumption stage.

2. The section 3.2 is a little bit messy as the authors should first claim the assumption and then show the MAE values. The current presentation may make the readers confused at first impression.

3. Please use notations to represent attention mass to prevent confusion.

I am glad to increase my score if my concerns are alleviated during the rebuttal stage.

References:\
[1] Anthropic. In-context Learning and Induction Heads. 2022.\
[2] Sun et al. Massive Activations in Large Language Models. COLM 2024.

**Questions:**

Please see the weakness.

---

> ### Author Response · Authors · 2025-11-19
> **Response to reviewer 9XZ2**
>
> Dear Reviewer,
>
> Thank you very much for your thoughtful review.
>
> Weaknesses:
> 1. Thank you for raising this point. We agree that, taken literally, an exponential approximation cannot capture truly negative cosine similarities. Our intention with Assumption 2 is to model the average positional decay trend of cross-token similarity, not to claim that every individual cosine is strictly non-negative. Occasional negative cosines do occur, but they are relatively rare and mostly reflect the behavior of the sink token.
> In the revision we will clarify this explicitly in the text and add an illustrative figure showing $\cos⁡(v[i],v[j])$ for, e.g., layer 10. As can be seen from this plot, some cosine values are negative, but the majority of these correspond to interactions involving the sink token.
>
> 2. Our goal with Assumption 3 is to model the typical large-scale shape of attention as a function of position: (i) a strong sink at the first token, (ii) a relatively flat “background’’ plateau, (iii) an oscillatory regime, and (iv) an overall recency bias. Figure 4 is computed as an average over heads, layers, and sequences, and is meant to illustrate this population-level trend, not to claim that every head follows this pattern exactly.
>
> We fully agree that there exist induction heads [1] and secondary sink-like tokens in the middle of the context [2]. These are not ruled out by Assumption 3:
>
>  – induction heads typically manifest as strong, localized peaks at specific offsets, which in our parametrization are captured inside the oscillatory phase (the cosine term can represent such structured peaks rather than a perfectly smooth wave);
>
>  – additional sink-like positions correspond to heads where the “sink’’ component is not confined to the first token only, but where multiple positions receive elevated, relatively position-invariant mass.
>
> Importantly, our theoretical bounds do not require that all heads admit a low-error fit to this piecewise profile. The proofs use only a small set of aggregate quantities derived from the profile (total sink mass, baseline mass, recency growth), which act as an envelope on the attention distribution.
>
> Regarding fitting errors, in the current version we chose to emphasize the average behavior of the attention weights rather than to demonstrate that the parametric law is an excellent pointwise approximation. Our goal was to propose a simple structure that captures the main regimes of the attention weights, not to optimize parameter estimation. Nevertheless, in the revised version, we will report MAE between the OLS-fitted profiles and the empirical weights to make the quality of fit more explicit.
>
> 3. Thank you for this observation. The theoretical bounds and derivations are formulated for general $L$, $N$, and other parameters, but in the empirical section we indeed focus primarily on the last-token query for clarity. When we switch to the multi-head analysis and examine the logic behind head geometry, we first determine head types using information across all positions $L$ (see Figure 10). We will clarify this more explicitly in the text and, where space permits, add summary statistics indicating that the main phenomena are not specific to the last token.
>
> 4. We will include the additional plots and statistics mentioned above (for example, examples of negative cosines and quantitative fit-quality measures) in the revised version to make the empirical support for our assumptions and parametrization clearer.
>
> 5. Thank you for the positive assessment of the theoretical framework. We agree that its practical implications should be stated more clearly. Concretely, our analysis yields three actionable insights for the LLM/attention community:
>
>     a. Geometry-aware sparse / top-$N$ attention. Theorems 1–2 and Fig. 5 show that non-trivial separability is maximized in a small-$N$ regime (typically $N \in [1,4]$) and degrades at intermediate $N$. This directly informs how aggressively one can sparsify attention and which heads are suitable for top-$N$/token-selection schemes.
>
>      b. Sink-aware diagnostics. We show that sink similarity systematically biases Recall (Fig. 6). This provides a principled way to flag Reset-like heads that are poor long-range retrievers (and thus good pruning candidates) and to design sink-aware interventions (e.g., regularization of sink norm/similarity).
>
>       c. Head-level taxonomy. Our Retriever / Mixer / Reset taxonomy (Fig. 7–8) offers a simple geometric criterion to label head roles, which can be used for interpretability, pruning/compression, and routing—for example, selecting Retriever heads for long-context tasks.
>
>
> Thank you again for your careful and helpful review.

---

### Official Review · Reviewer_Ej41 · 2025-11-01

**Soundness:** 3
**Presentation:** 3
**Contribution:** 2
**Rating:** 6
**Confidence:** 4

**Summary:**

This paper takes a geometric look at how multi-head attention selects tokens.
The authors treat attention as a kind of classifier that operates in the value–state space and define geometric versions of Precision, Recall, and F-score to describe how well selected tokens are separated from the rest.
They assume three main conditions: (1) stable value norms and reduced sink activity, (2) exponential decay of cross-token similarity, and (3) a piecewise attention weight pattern that captures plateau–oscillation–recency behavior.
Under these assumptions, they derive non-asymptotic bounds on token separability and show that the best separation appears when only a few tokens (around 1–4) are selected.
Experiments on LLaMA-2-7B, Gemma-7B, and Mistral-7B back up these results and reveal three types of attention heads (Retriever, Mixer, and Reset) that differ in how they interact with the sink and final tokens.

**Strengths:**

1) The geometric framing is clear and easy to follow, giving an intuitive picture of how attention works.
2) The theory lines up closely with the empirical data across different models.
3) The head taxonomy (Retriever, Mixer, Reset) offers a concrete and interpretable way to describe functional differences between heads.
4) The paper doesn’t rely on extra training or architectural changes, making the analysis generally applicable.

**Weaknesses:**

(1) Limited practical connection.
The paper gives a clean geometric description of token selection and supports it with strong empirical evidence.
However, it stops short of connecting these findings to model performance or design improvements.
The results are insightful but remain mainly diagnostic, without demonstrating benefits such as better alignment, loss reduction, or architectural efficiency.

(2) Assumption sensitivity.
The theoretical derivations rely on several empirical assumptions — stable value norms, exponential similarity decay, and piecewise attention profiles.
These assumptions are plausible but not formally justified, and the paper does not examine cases where they fail (e.g., fine-tuned models, longer contexts, or high-variance heads).
The robustness of the geometric bounds under such conditions remains unclear.

(3) Missing broader comparison.
The analysis focuses entirely on top-N token selection and does not compare against more continuous or weighted formulations of attention.
As a result, it is uncertain whether the observed separability patterns are specific to discrete selection or general to the full attention mechanism.

**Questions:**

(1) Can the proposed geometric Precision/Recall or F-score metrics be linked to downstream performance (e.g., loss, perplexity, or alignment quality)?

(2) How sensitive are the theoretical bounds to violations of the key assumptions (norm stability, similarity decay, or piecewise attention profiles)?

(3) Could the geometric interpretation be extended into token pruning or head sparsification methods in practice?

(4) Do the Retriever, Mixer, and Reset head types appear consistently across different model sizes, architectures, or data domains?

(5) Both this paper and Orthorank(Shin et al., 2025) report a similar phenomenon, stable norms except for the sink token. Are these two observations fundamentally related, or do they arise independently in different representational spaces (hidden vs. value)?

---

> ### Author Response · Authors · 2025-11-19
> **Response to reviewer Ej41**
>
> Dear Reviewer,
> Thank you for your thoughtful and constructive review.
>
> Weaknesses
> 1.  We partially agree with your comment. The goal of our work was to theoretically examine the geometric properties of classical Transformers. While we fully acknowledge that a practical investigation is important, establishing a complete theoretical connection between model performance and its geometric characteristics is, in our view, substantial enough to require a separate paper.
> 2. Thank you for raising this point. We attempted to justify our assumptions using empirical evidence (in particular, the analysis of Llama-7B parameters) and showed that in more than 90% of heads these assumptions hold. However, we agree that certain head-layer pairs behave differently from what our assumptions predict, leading to more specific behaviors that are not fully captured by our current framework. We will clarify this limitation more explicitly in the revised version.
> 3. Our focus was on attention as an information selector, which is why we analyzed top-$N$ selection. It is indeed an interesting question what would happen in a fully continuous formulation of the problem. However, similar to point 1, we believe that such an extension would be better treated in a separate work, rather than in a paper that aims primarily to analyze the classical architecture.
>
> Questions
>
> 1. We believe your statement is correct. Although there is currently no established theoretical framework directly linking model performance metrics to geometric metrics, there are several relevant connections. First, entropy properties of attention are related to its geometric behavior: the distribution of vectors and their cross-relations are closely connected to the weights in the attention matrix (as illustrated in the figures in our paper). Moreover, prior work (see, e.g., https://arxiv.org/abs/2412.16545) indicates a link between entropy properties and model performance.
> Second, we show that different head regimes correspond to different Precision/Recall behaviors; therefore, different heads implement different “logics.” In https://arxiv.org/abs/1905.10650, the authors demonstrate that some heads can be removed without loss in performance. We expect that a future, more systematic analysis of our head classification could suggest how to re-aggregate or prune heads to make models faster while maintaining performance.
>
> 2. From a theoretical point of view, the crucial assumptions are (i) stability of the norm and (ii) similarity of value states. These assumptions allow us to simplify the analysis and derive informative bounds. The resulting theoretical bounds are relatively robust to changes in other assumptions. We believe it is possible to relax these assumptions to make them more realistic, but this would significantly complicate the theoretical part and make it less transparent.
>
> 3. Yes, especially in the context of head sparsification. Our analysis aligns with the idea that not all heads are informative (both theoretically and empirically). We observe that different head regimes exhibit distinct internal geometries, which is consistent with the sparsification perspective.
>
> 4. Yes. We did not include the distribution of head types in the current version, but we will add it in the revision. In general, the pattern is roughly as follows: the early layers show more diversity, followed by a tendency toward full reset, and then a re-accumulation of the Retriever–Mixer–Reset distribution.
>
> 5. We believe these observations are indeed theoretically connected. Since value vectors are linear projections of hidden states, stability of hidden-state norms naturally induces stability of norms in value space under well-behaved projections.
>
> Thank you again for your careful reading and helpful comments. We hope our responses address your concerns and clarify our choices.

---

### Meta-Review · Area_Chair_JaYh · 2025-12-30

**Summary:**

This paper studies how multi-head attention selects tokens by taking a geometric viewpoint. To do so, the authors give geometric definitions of Precision, Recall and F-score and, under three assumptions (stable value norms and reduced sink activity; exponential decay of cosine similarity between tokens; a certain attention weight pattern), they provide non-asymptotic bounds on them. The theoretical results are validated via an experimental study on 7B LLMs, which showcase different types of attention heads.

On the positive side, all reviewers appreciated the novelty of the geometrical framing (and so do I). On the negative side, all reviewers noted the limited practical insights coming from the findings. Other issues that have been pointed out are the relatively limited experimental depth and the strength of the assumptions. Unfortunately, I agree that these are relatively major shortcomings which are not fully addressed by the rebuttal. Thus, I recommend a rejection at this stage. As the idea is interesting and the manuscript does have potential, I do encourage the authors to submit a revised version to a future venue.

**Reviewer Concerns:**

The thoughtful rebuttal of the authors addressed several concerns, in particular:
* Weakness 3, question 3 and question 5 of reviewer Ej41
* Weaknesses 1, 2, 3 of reviewer 9XZ2
* Weaknesses 1, 2, 4 and questions 1-2 of reviewer uLa2

Other issues though remain outstanding and they would require a major revision:
* Weaknesses 1-2 of reviewer Ej41
* Weaknesses 4-5 of reviewer 9XZ2
* Weaknesses 3, 5, 6 of reviewer uLa2

**Reviewer Scores:**

My best guess is that, after a full discussion, the changes in the scores (and general opinions of the reviewers) would not have been sufficient for the paper to meet the bar for acceptance.

---

### Decision · Program_Chairs · 2026-01-26

Reject